# Nonlinear Charged Black Hole Solution in Rastall Gravity

Gamal Gergess Lamee Nashed

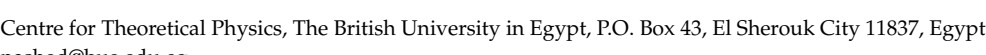

Centre for Theoretical Physics, The British University in Egypt, P.O. Box 43, El Sherouk City 11837, Egypt;
nashed@bue.edu.eg

**Abstract:** We show that the spherically symmetric black hole (BH) solution of a charged (linear case) field equation of Rastall gravitational theory is not affected by the Rastall parameter and this is consistent with the results presented in the literature. However, when we apply the field equation of Rastall's theory to a special form of nonlinear electrodynamics (NED) source, we derive a novel spherically symmetric BH solution that involves the Rastall parameter. The main source of the appearance of this parameter is the trace part of the NED source, which has a non-vanishing value, unlike the linear charged field equation. We show that the new BH solution is Anti−de-Sitter Reissner−Nordström spacetime in which the Rastall parameter is absorbed into the cosmological constant. This solution coincides with Reissner−Nordström solution in the GR limit, i.e., when Rastall's parameter is vanishing. To gain more insight into this BH, we study the stability using the deviation of geodesic equations to derive the stability condition. Moreover, we explain the thermodynamic properties of this BH and show that it is stable, unlike the linear charged case that has a second-order phase transition. Finally, we prove the validity of the first law of thermodynamics.

**Keywords:** Rastall gravitational theory; black hole; thermodynamics and first law





## 1. Introduction

Since the construction of Einstein's general relativity (GR), the coupling between a scalar field and the gravitational action in a geometric frame has been intensively studied. A scalar theory formulation was made in [1], and Jordan–Brans–Dicke later built a gravitational theory as an expansion of GR to investigate the variable of gravitational coupling [2–4]. Afterward, a general combination between a scalar field and its derivative, which yields second-order differential equations, is known as the Horndeski theory [5] that gained much attention. Recently, many modifications of Einstein GR have been established. Among these theories is the $f(R)$ gravitational theory, which is regarded as a natural generalization of Einstein's Hilbert action [6]. This theory could be rewritten as a GR and scalar field [7,8]. The above is a very brief summary related to the scalar fields in the frame of a gravitational context. However, there is a huge literature on this subject.

The above discussions show one way of modification of GR. However, there is another possibility that has been used to generalize the kinetic term of the scalar field that is minimally coupled to the Einstein–Hilbert action. This possibility is called the k-essence theory [9]. This theory is used as an option to the usual inflationary models that use a self-interacting scalar field [9–14]. Recently, vacuum static spherically symmetric solutions have been derived for the k-essence theories [14]. Some novel patterns have been derived that involve a study of the event horizon. Nevertheless, interpolating such solutions as black holes was difficult because it is impossible to define a distant region from the horizon. Using the no-go theorem, it has been affirmed that solutions with a regular horizon can exist but only of the type of cold black hole [15,16].

Another generalization of GR is to abound the restriction of the conservation law encoded in the zero divergence of the energy-momentum tensor. Among the theories that follow this direction is the one given by Rastall (1972), which is known as Rastall's theory [17]. In the frame of Rastall theory, the covariant divergence of the stress-energy

momentum tensor is proportional to the covariant divergence of the curvature scalar, i.e., $T^{\alpha}{}_{\beta;\alpha} \propto R_{;\beta}$. Thus, any solution that has a zero or constant Ricci scalar Rastall theory will be identical to Einstein GR. Explaining the behavior of the new source of Rastall's theory is not an easy task. We can consider, phenomenologically, this new source as an appearance of quantum effects in the classical frame [18]. It is interesting to mention that the topic of non-conservation of $T^{\alpha\beta}$ is a feature that exists in diffusion models [19–23]. Furthermore, the non-conservation of the energy-momentum tensor and its link to modified gravitational theories has been analyzed in [24,25]. The variational principle in the frame of Riemannian geometry is not held due to the non-conservation of $T^{\alpha\beta}$. Nevertheless, some features such as Rastall's theory can also be discovered in the frame of Weyl geometry [26]. Moreover, external fields in the Lagrangian could give essentially the same behavior as Rastall's theory (for discussion of the external field see, for example [27]). An investigation of Rastall gravity, for an anisotropic star with a static spherical symmetry, has been discussed in [28]. The study of shadow and energy emission rates for a spherically symmetric non-commutative black hole in Rastall gravity has been carried out [29]. The quasinormal modes of black holes in Rastall gravity in the presence of non-linear electrodynamic sources have been studied [30]. Moreover, the quasinormal modes of the massless Dirac field for charged black holes in Rastall gravity have been discussed [31]. In the framework of Rastall gravity, a new black hole solution of the Ayón-Beato-García type, surrounded by a cloud of strings, is derived [32]. A solution of a static spherically symmetric black hole surrounded by a cloud of strings in the frame of Rastall gravity is derived [33]. Moreover, two classes of black hole (BH) solutions, conformally flat and non-singular BHs, are presented in [34]. A spherically symmetric gravitational collapse of a homogeneous perfect fluid in Rastall gravity has been conducted in [35]. Oliveira also presented static and spherically symmetric solutions for the Rastall modification of gravity to describe neutron stars [36].

In the frame of cosmology, Rastall's theory could degenerate into the $\Lambda$ cosmological dark matter, $\Lambda$CDM, at the background and at first-order levels, which means that a viable model can be constructed in the frame of this theory. However, a few applications in the domain of astrophysics have been completed [37]. Additionally, a study of the generalized Chaplygin gas model to fit observations has been carried out in Rastall theory [38]. The quantum thermodynamics of the Schwarzschild-like black hole found in the bumblebee gravity model has been discussed in [39]. In recent years, various BH solutions, and in particular, BH solutions of the Rastall field equations, have been investigated in many scientific research papers. Among these are charged static spherically symmetric BH solutions [40,41], Gaussian BH solutions [42,43], rotating BH solutions [44,45], Abelian–Higgs strings [46], Gödel-type BH solutions [47], black branes [48], wormholes [49], BH solutions surrounded by fluid, electromagnetic field [50] or quintessence fluid [51], BH thermodynamics [52], among other theoretical efforts [53–56]. It is the aim of the present study to show the effect of the Rastall parameter in the domain of spherically symmetric spacetime using a special form of NED coupled with Einstein's GR.

This paper has the following structure: in the next section, we present a summary of Rastall's theory. In Section 2.1, we give the NED field equations of Rastall's gravity, then we apply them to a spherically symmetric spacetime with two unequal metric potentials and derive the NED differential equations. We solve this system and derive a new BH solution that involves Rastall's parameter. In Section 2.2, we extract the physical properties of the BH solution and show that the metric potentials asymptote as Anti-de-Sitter (A)dS Reissner–Nordström. Despite the applied NED field equations without cosmological constant, we obtain (A)dS Reissner–Nordström. This means that the Rastall parameter acts as a cosmological constant in this special form of NED theory. This result is consistent with the study given by Visser [57]. It is important to stress that this solution in the GR limit, i.e., when the Rastall parameter equals zero, coincides with the Reissner−Nordström solution. In Section 2.3, we derive the stability of geodesic motion using geodesic deviations. In Section 3, we study some thermodynamical quantities. In Section 3.1, we show that our BH

satisfies the first law of thermodynamics. In Section 4, we discuss the output results of this study.

## 2. Spherically Symmetric BH Solution

Rastall's assumptions [17,58], for a spacetime with a Ricci scalar $R$ filled by an energy-momentum $T_{\mu\nu}$, we have:

$$T^{\alpha\beta}{}_{;\alpha} = \epsilon \mathcal{R}_;{}^{\beta}, \tag{1}$$

where $\epsilon$ is the Rastall parameter, which is responsible for the deviation from the standard GR conservation law. Equation (1) returns to Einstein's GR when the Ricci scalar is vanishing or has a constant value.

Using the above data, we can write the Rastall field equations in the form [17,58]:

$$\mathcal{R}_{\alpha\beta} - \left[\frac{1}{2} + \lambda\right] g_{\alpha\beta}\mathcal{R} = \chi T_{\alpha\beta}, \tag{2}$$

where $\lambda = \chi\epsilon$ and $\chi$ is the Newtonian gravitational constant and the units are used so that the speed of light $c = 1$. Here, $\mathcal{R}_{\alpha\beta}$ is the Ricci tensor, $\mathcal{R}$ is the Ricci scalar, $g_{\alpha\beta}$ is the metric tensor, and $T_{\alpha\beta}$ is the energy-momentum tensor describing the material content. The constant $\epsilon$ is the Rastall parameter that is responsible for the deviation from GR and when ($\epsilon = 0$) we obtain GR theory.

The modification in the spacetime geometry given by the L.H.S. of Equation (2) links to two modifications of different material contents of the right hand side of Equation (2):

(i) Firstly, Equation (2) is mathematically equivalent to adding new materials of the actual material sources to the right hand side of the standard GR field equations, which can be seen as an effective source accompanying the actual material sources considered in the model. For this reason, we can rewrite Equation (2) in a mathematical equivalent form as [17,58]:

$$\mathcal{R}_{\alpha\beta} - \frac{1}{2}g_{\alpha\beta}\mathcal{R} = \chi T_{1\alpha\beta}, \quad \text{where} \quad T_{1\alpha\beta} = T_{\alpha\beta} - \frac{\chi\epsilon}{1+4\chi\epsilon}g_{\alpha\beta}T. \tag{3}$$

The term $-\frac{\epsilon}{1+4\epsilon}g_{\alpha\beta}T$ is the energy-momentum tensor that represents the effective source that arises from the actual material and $T$ is the trace of $T_{\alpha\beta}$, i.e., $T = g_{\alpha\beta}T^{\alpha\beta} = -(1+4\epsilon)R$. Now rewrite Equation (3) in the form: [1]

$$\mathcal{R}_{\alpha\beta} - g_{\alpha\beta}\mathcal{R}\left[\frac{1}{2} + \epsilon\right] \equiv \mathcal{R}_{\alpha\beta} + g_{\alpha\beta}T\left[\frac{1+2\epsilon}{2(1+4\epsilon)}\right] = T_{\alpha\beta}. \tag{4}$$

In this study, we will use Equation (4) but we will assume the energy-momentum tensor $T_{\alpha\beta}$ to be combined with electromagnetic field and takes the following form:

$$T_{\alpha\beta} = E_{\alpha\beta}, \quad \text{where} \quad E_{\alpha\beta} = F^{\mu}{}_{\alpha}F_{\mu\beta} - \frac{1}{4}g_{\alpha\beta}F, \tag{5}$$

with $F_{\mu\beta}$ being the antisymmetric Faraday tensor and $F = F^{\mu\nu}F_{\mu\nu} = d\xi$ and $\xi = \xi_{\alpha}dx^{\alpha}$ is the electromagnetic gauge potential Maxwell field [59]. The tensor $F_{\mu\beta}$ satisfies the vacuum Maxwell equations:

$$F^{\alpha\beta}{}_{;\alpha} = 0, \quad F_{\alpha\beta;\sigma} + F_{\beta\sigma;\alpha} + F_{\sigma\alpha;\beta} = 0. \tag{6}$$

(ii) Secondly, this modification implies a violation of the local conservation of the tensor $T_{1\alpha\beta}$ of an actual material source because its divergence is not necessarily vanishing.

It is important to stress that Equation (4) with the energy-momentum tensor given by Equation (6) has a contradiction since the LHS of Equation (4) has a non-vanishing covariant derivative, $\left\{\mathcal{R}^{\alpha\beta} - g^{\alpha\beta}\mathcal{R}\left[\frac{1}{2} + \frac{\epsilon}{1+4\epsilon}\right]\right\}_{;\beta} \neq 0$, while the RHS has a vanishing

value, $T^{\alpha\beta}{}_{;\beta} = 0$. Thus, the only way to overcome this issue is the fact that the solution of these field equations must have a zero Ricci scalar[2], which ensures the well-known results in the literature that the Rastall parameter has no effect in the linear Maxwell field.

### 2.1. Nonlinear Charged Spherically Symmetric BH Solution in Rastall's Theory

In this subsection, we are going to present a special form of NED theory coupled with GR. For this aim, we are going to take into account a dual representation, i.e., imposing the auxiliary field $\mathcal{S}_{\alpha\beta}$, which is convenient to couple with GR [60,61]. Specifically, we involve the Legendre transformation:

$$\mathbb{H} = 2FL_F - L, \tag{7}$$

where $\mathbb{H}$ is an arbitrary function, $L_F \equiv \frac{\partial L}{\partial F}$ and $L(F)$ is an arbitrary function of $F$. If $L(F) = F$ we return to the linear case. Assuming,

$$\mathcal{S}_{\mu\nu} = L_F F_{\mu\nu}, \qquad \mathcal{S} = \frac{1}{4}\mathcal{S}_{\alpha\beta}\mathcal{S}^{\alpha\beta} = L_F^2 F, \qquad \text{with} \qquad F_{\mu\nu} = \mathbb{H}_{\mathcal{S}}\mathcal{S}_{\mu\nu}, \tag{8}$$

where $\mathbb{H}_{\mathcal{S}} = \frac{\partial \mathbb{H}}{\partial \mathcal{S}}$. The field equation of nonlinear electrodynamics yields the form [60]:

$$\partial_\nu\left(\sqrt{-g}\mathcal{S}^{\mu\nu}\right) = 0, \tag{9}$$

where the energy-momentum tensor of the NED is defined as:

$$T^\nu{}_\mu{}^{NED} \equiv 2(\mathbb{H}_{\mathcal{S}}S_{\mu\alpha}S^{\nu\alpha} - \delta^\nu_\mu[2S\mathbb{H}_{\mathcal{S}} - \mathbb{H}]). \tag{10}$$

We mention that in general Equation (10) has a non-vanishing trace[3]:

$$T^{NED} = 8(\mathbb{H} - \mathbb{H}_{\mathcal{S}}\mathcal{S}) \neq 0, \tag{11}$$

and has a vanishing value in the linear theory, i.e., when $\mathbb{H} = F$ and $\mathcal{S} = F$. Finally, the electric and magnetic fields in the NED case take the form [60,61]:

$$E = \int F_{tr}dr = \int \mathbb{H}_S S_{tr}dr, \qquad B_r = \int F_{r\phi}d\phi = \int \mathbb{H}_S S_{r\phi}d\phi,$$

$$B_\theta = \int F_{\theta r}dr = \int \mathbb{H}_S S_{\theta r}dr, \qquad B_\phi = \int F_{\phi r}dr = \int \mathbb{H}_S S_{\phi r}dr, \tag{12}$$

where $E$ and B are the components of the electric and magnetic fields, respectively. Now we are going to use the field Equation (4) with the energy-momentum tensor $T_{\alpha\beta}$, that is combined with the NED, and obtain:

$$T_{\mu\nu}{}^{NED} \equiv \mathbb{E}_{\mu\nu}, \qquad \text{where} \qquad \mathbb{E}_\mu{}^\nu = 2(\mathbb{H}_S S_{\mu\alpha}S^{\nu\alpha} - \delta^\nu_\mu[2S\mathbb{H}_S - \mathbb{H}]). \tag{13}$$

Now, let us assume that the spherically symmetric spacetime has the form:

$$ds^2 = -\mu(r)dt^2 + \frac{dr^2}{\nu(r)} + r^2(d\theta^2 + \sin^2\theta\, d\phi^2), \tag{14}$$

where $\mu(r)$ and $\nu(r)$ are unknown functions of the radial coordinate $r$. For the spacetime (14), the symmetric affine connection takes the form:

$$\Gamma_{tt}{}^r = \frac{1}{2}\nu\mu', \qquad \Gamma_{tr}{}^t = \frac{\mu'}{2\mu}, \qquad \Gamma_{rr}{}^r = \frac{\nu'}{2\nu}, \qquad \Gamma_{r\theta}{}^\theta = \Gamma_{r\phi}{}^\phi = \frac{1}{r},$$

$$\Gamma_{\theta\theta}{}^r = -r\nu, \qquad \Gamma_{\theta\phi}{}^\phi = \cot\theta, \qquad \Gamma_{\phi\phi}{}^r = -r\nu\sin^2\theta, \qquad \Gamma_{\phi\phi}{}^\theta = -\sin\theta\cos\theta. \tag{15}$$

The Ricci scalar of the spacetime (14) has the form:

$$\mathrm{R(r)} = \frac{r^2 v\mu'^2 - r^2\mu\mu'v' - 2r^2\mu v\mu'' - 4r\mu[v\mu' - \mu v'] + 4\mu^2(1 - v)}{2r^2\mu^2}.$$ (16)

Here, $\mu \equiv \mu(r)$, $v \equiv v(r)$, $\mu' \equiv \frac{d\mu}{dr}$, $\mu'' \equiv \frac{d^2\mu}{dr^2}$ and $v' \equiv \frac{dv}{dr}$.

Using Equation (14) in Equation (4), where the energy-momentum tensor is given by Equation (13), then:

The $t\,t$- component of Rastall field equation is:

$$\frac{1}{2r^2\mu^2[2\mu v\xi'' + \xi'(\mu v' - v\mu')]}\Bigg\{2\mu v\xi''\Big\{2\mu v\epsilon r^2\mu'' - r^2 v\epsilon\mu'^2 + r\mu\mu'\epsilon[4v + rv'] + 2\mu^2[(1 + 2\epsilon)[rv' + v] + 1 + r^2\mathbb{H} + 2\epsilon]\Big\}$$

$$+\xi'\Big(2r^2\mu v\mu''\epsilon[\mu v' - v\mu'] + r^2v^2\epsilon\mu'^3 - 2r\mu v\mu'^2[rv' + 2v]\epsilon + \mu^2\mu'\big[r^2\epsilon v'^2 - 2rvv' - 2v(\{1 + 2\epsilon\}(v - 1) - r^2\mathbb{H})\big]$$

$$+\mu^3\Big\{2v'[(1 + 2\epsilon)\{rv' + v - 1\} - r^2\mathbb{H}] + r^2v\mathbb{H}'\Big\}\Big)\Bigg\} = 0,$$

The $r\,r$- component of Rastall field equation is:

$$\frac{1}{2r^2\mu^2[2\mu v\xi'' + \xi'(\mu v' - v\mu')]}\Bigg\{2\mu v\xi''\Big\{2\mu v\epsilon r^2\mu'' - r^2 v\epsilon\mu'^2 + r\mu\mu'[2(1 + 2\epsilon)v + \epsilon rv'] + 2\mu^2[(1 + 2\epsilon)[v - 1] + 2\epsilon rv' - r^2\mathbb{H}]\Big\}$$

$$+\xi'\Big(2r^2\mu v\mu''\epsilon[\mu v' - v\mu'] + r^2v^2\epsilon\mu'^3 - 2r\mu v\mu'^2[rv'\epsilon + (1 + 2\epsilon)v] + \mu^2\mu'\big[r^2\epsilon v'^2 + 2rvv' - 2v(\{1 + 2\epsilon\}(v - 1) - r^2\mathbb{H})\big]$$

$$+2\mu^3\Big\{2\epsilon rv'^2 + v'[(1 + 2\epsilon)\{v - 1\} - r^2\mathbb{H}] + r^2v\mathbb{H}'\Big\}\Big)\Bigg\} = 0,$$

The $\theta\,\theta = \phi\,\phi$- component of Rastall field equation is:

$$\frac{1}{4r^2\mu^2[2\mu v\xi'' + \xi'(\mu v' - v\mu')]}\Bigg\{2\mu v\xi''\Big\{2\mu v(1 + 2\epsilon)r^2\mu'' - r^2 v(1 + 2\epsilon)\mu'^2 + r\mu\mu'[2(1 + 4\epsilon)v + (1 + 2\epsilon)rv'] + 2\mu^2[4\epsilon[v - 1]$$

$$+(1 + 4\epsilon)rv' - 2r^2\mathbb{H}]\Big\} + \xi'\Big(2r^2\mu v\mu''(1 + 2\epsilon)[\mu v' - v\mu'] + r^2v^2(1 + 2\epsilon)\mu'^3 - 2r\mu v\mu'^2[rv'(1 + 2\epsilon) + (1 + 4\epsilon)v]$$

$$+\mu^2\mu'\big[r^2(1 + 2\epsilon)v'^2 - 4v(2\epsilon(v - 1) - r^2\mathbb{H})\big] + 2\mu^3\Big\{(1 + 4\epsilon)rv'^2 + 2v'[2\epsilon\{v - 1\} - r^2\mathbb{H}] + 4r^2v\mathbb{H}'\Big\}\Big)\Bigg\} = 0,$$ (17)

where $\mathbb{H}$ is an arbitrary function and $\xi$ is the field of electric charge. Equation (17) reduces to the linear charged Einstein's field equations when $\epsilon = 0$ and $\mathbb{H} = F$ [62,63]. The exact solution of Equation (17) for the electric field takes the form[4]:

$$\mu(r) = \frac{c_2(c_3r^4 + (1 + 4\epsilon)[12r^2 + 12rc_4 - 3c_5])}{r^2}, \qquad v(r) = \frac{c_3r^4 + (1 + 4\epsilon)[12r^2 + 12rc_4 - 3c_5]}{12r^2(1 + 4\epsilon)},$$

$$\xi(r) = \frac{c_1}{r}, \qquad \mathbb{H} = c_3 + \frac{c_5}{r^4} \equiv c_3 + F.$$ (18)

The Rastall parameter has an effect in the NED case, as shown by Equation (18). We return to the linear charged case when $\mathbb{H} = F \equiv \frac{c_5}{r^4}$ [64]. We stress the fact that if we repeat the same above calculations taking into account the electric and magnetic fields, given by Equation (12), we can easily verify the same conclusion of the above case, i.e., Rastall's parameter has an effect and its behavior will be similar to the form given by Equation (18). If we want to derive a solution that is different from Einstein's GR, we must generalize Rastall's theory to $f(R)$-Rastall's theory [65]

### 2.2. The Physical Properties of the BH Solutions (18)

Now, we are going to explain the physics of the BH solution (18). For such purposes, we rewrite the components of the metric potential of the BH (18) as:

$$\mu(r) = \nu(r) = r^2 \Lambda_{eff} + 1 - \frac{2M}{r} + \frac{q^2}{r^2}, \qquad \xi = -\frac{q}{r}, \qquad \mathbb{H} = 12\Lambda_{eff.}(1 + 4\epsilon) - \frac{4q^2}{r^4}. \tag{19}$$

where we have:

$$c_1 = -q, \quad c_2 = \frac{1}{12(1 + 4\epsilon)}, \quad \Lambda_{eff.} = c_3 c_2, \quad c_4 = -2M, \quad \text{and} \quad -4q^2 = c_5, \quad c_5 = -4\sqrt{c_1}. \tag{20}$$

Equation (20) shows that we have an effective cosmological constant in the solution of the NED charged case while their field equations have no cosmological constant. This means that the Rastall parameter acts as an effective cosmological constant in the NED charged case with the fact that the Rastall parameter $\epsilon \neq -\frac{1}{4}$. From Equations (19) and (14) we obtain[5]:

$$ds^2 = -\left\{ r^2 \Lambda_{eff} + 1 - \frac{2M}{r} + \frac{q^2}{r^2} \right\} dt^2 + \frac{dr^2}{r^2 \Lambda_{eff} + 1 - \frac{2M}{r} + \frac{q^2}{r^2}} + d\Omega^2, \tag{21}$$

where $d\Omega^2 = r^2(d\theta^2 + \sin^2\theta \, d\phi^2)$ is a 2-dimensional unit sphere.

Equation (21) shows that solution (18) asymptotes as (A)dS and does not equal Reissner–Nordström spacetime due to the Rastall parameter. Equation (21) clearly investigates how the Rastall parameter acts as a cosmological constant. Equation (21) coincides with GR when $\mathbb{H} = F$, which means $c_3 = 0$, and this gives Rissner–Nordström BH solution because $\Lambda_{eff} = 0$. From Equations (19) and (16) we achieve:

$$R(r) = -12\Lambda_{\text{eff}}. \tag{22}$$

Equation (22), shows in a clear way that the Rastall parameter acts as a cosmological constant and the conservation law of both sides of Equation (2) are satisfied.

Using Equation (19) we obtain the invariants as:

$$\mathcal{R}_{\mu\nu\rho\sigma}\mathcal{R}^{\mu\nu\rho\sigma} = -24\Lambda_{\text{eff}} + \frac{48m^2}{r^6} - \frac{96mq^2}{r^7} + \frac{56q^4}{r^8}, \qquad \mathcal{R}_{\mu\nu}\mathcal{R}^{\mu\nu} = 36\Lambda_{\text{eff}} + \frac{4q^4}{r^8}, \qquad \mathcal{R} = -12\Lambda_{\text{eff}}. \tag{23}$$

Here $\left(\mathcal{R}_{\mu\nu\rho\sigma}\mathcal{R}^{\mu\nu\rho\sigma}, \mathcal{R}_{\mu\nu}\mathcal{R}^{\mu\nu}, \mathcal{R}\right)$ are the Kretschmann scalar, the Ricci tensor square, and the Ricci scalar, respectively. The Kretschmann scalar and the Ricci tensor square have a true singularity when $r = 0$. All of the above invariants are identical with the invariant of $(A)dS$-Reissner–Nordström BH solution of GR. The discussion of the invariant of (A)dS Reissner–Nordström can be applied on the invariant given by Equation (23) with the exclusion of the value $\epsilon = -\frac{1}{4}$.

Before we close this subsection we are going to calculate the trace of the NED given by Equation (11) using solution (18) as:

$$T^{NED} = c_3 \neq 0. \tag{24}$$

Equation (24) shows in a clear way that if $c_3 = 0$ we will obtain a vanishing trace and, in that case, Rastall's parameter will have no effect, which supports the above discussion.

### 2.3. Stability of Geodesic Motion of BH Given by Equation (19)

The equations of geodesic are given by [66]:

$$\frac{d^2 x^\gamma}{d\varepsilon^2} + \left\{ \begin{array}{c} \gamma \\ \beta\rho \end{array} \right\} \frac{dx^\beta}{d\varepsilon} \frac{dx^\rho}{d\varepsilon} = 0, \tag{25}$$

where $\varepsilon$ is a canonical parameter. Moreover, the equations of geodesic deviation are given as [67,68]:

$$\frac{d^2\varrho^\sigma}{d\varepsilon^2} + 2\begin{Bmatrix} \sigma \\ \mu\nu \end{Bmatrix}\frac{dx^\mu}{d\varepsilon}\frac{d\varrho^\nu}{d\varepsilon} + \begin{Bmatrix} \sigma \\ \mu\nu \end{Bmatrix}_{,\rho}\frac{dx^\mu}{d\varepsilon}\frac{dx^\nu}{d\varepsilon}\varrho^\rho = 0, \tag{26}$$

where $\varrho^\alpha$ is the deviation of the four-vector.

Following the procedure in [69,70], one can get the stability condition as:

$$\frac{3\mu\nu\mu' - \sigma^2\mu\mu' - 2r\nu\mu'^2 + r\mu\nu\mu''}{\mu\nu'} > 0, \tag{27}$$

where $\mu$ and $\nu$ are given by Equation (19). Using Equation (27), one can obtain the following form of $\sigma^2$ as:

$$\sigma^2 = \frac{3\mu\nu\mu'' - 2r\nu\mu'^2 + r\mu\nu\mu''}{\mu^2\nu'^2} > 0. \tag{28}$$

Equation (28) is plotted in Figure 1 using specific values of the model. In this figure, we study $\Lambda_{eff} = 0$, Reissner−Nordström GR spacetime and $\Lambda_{eff} \neq 0$ of the BH solution (19). The two cases display the regions where the BH solution is stable/unstable by unshaded and shaded regions, respectively.

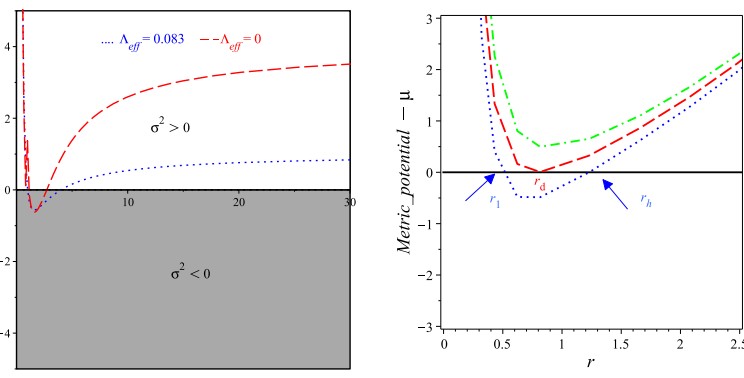

(a) Stability of the BH for the case $\Lambda_{eff} = 0$ and $\Lambda_{eff.} = 0.083$

(b) Horizons of the linear Maxwell field

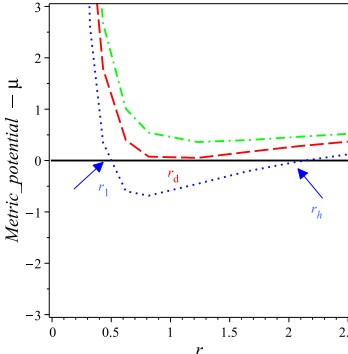

(c) Horizons of the non–linear Maxwell field

**Figure 1.** Plot (**a**) shows the behavior of Equation (28) viz $r$ for BH (19). The behavior of the metric potential $\mu(r)$, which characterizes the horizons by putting $\mu(r) = 0$: (**b**) for linear Maxwell Rastall gravity theory; (**c**) for the nonlinear electrodynamics Rastall's theory. The values of $m$ for the linear case are 1.3; 0.99; 0.8 and q = 1, while for the nonlinear case m = 1.3; 1.1 and 0.9, q = 1 and $\Lambda_{eff.} = 0.3$.

### 3. The Thermodynamical Properties of the of BH Given by Equation (19)

The thermodynamics of BH is considered an interesting topic in physics because it enables us to understand the physics of the solution. Two main approaches have been proposed to understand the thermodynamical quantities of the BHs: The first approach, delivered by Gibbons and Hawking [71,72] constructed to understand the thermal properties of the Schwarzschild BH through the use of Euclidean continuation. In the second approach, one has to define the gravitational surface from which we can define the Hawking temperature. Then, one can be able to study the stability of the BH [73–76]. Here, we are going to follow the second approach to investigate the thermodynamics of the (A)dS BH obtained in Equation (19) and then analyze its stability. The physical quantities characterized by the BH (19) are the mass, $m$, the charge, and the effective cosmological constant $\Lambda_{eff.}$.

The horizons of Equation (19) are calculated by deriving the roots of $\mu(r) = 0$, which we plot in Figure 1b,c using specific values. Plots of Figure 1b,c indicate the roots of $\mu(r)$ that fix the horizons of BH (19), i.e., $r_1$ and $r_h$ . We should emphasize that in the linear case, for $m > 0$, $q > 0$, and $\Lambda_{eff.} = 0$, we can show that the two roots can be formed when $m > m_{min} > q$. However, when $m = m_{min}$, we fix the degenerate horizons, i.e., $r_{dg}$, at which $r_1 = r_h$, which is the Nariai BH whose thermodynamics is studied [77–79]. However, when $m < m_{min} < q$, there is no BH formed, which means that we have a naked singularity as shown in Figure 1b. The same discussion can be used for the NED case, where the degenerate horizon is shown in Figure 1c [78–87]. In this study, we use positive values of the effective cosmological constant because this gives two horizons. Nevertheless, it is important to mention that negative values of the effective cosmological constant create the same pattern, which is characterized by two horizons [88,89]. The stability of the BH depends on the sign of the heat capacity $H_c$. Now, we are going to discuss the thermal stability of the BHs through their behavior of heat capacities [10,90–92]:

$$H_c = \frac{dE_h}{dT_h} = \frac{\partial m}{\partial r_h}\left(\frac{\partial T}{\partial r_h}\right)^{-1},\tag{29}$$

where $E_h$ is the energy. If $H_c > 0$ or ($H_c < 0$), the BH will be thermodynamically stable or unstable, respectively. To understand this process, we suppose that at some point the BH absorbs more radiation than it emits, which yields positive heat capacity, which means that the mass is indefinitely increased. In contrast, when the BH emits more radiation than it absorbs, this yields a negative heat capacity, which means the BH mass is indefinitely decreasing until it disappears. Therefore, a BH that has a negative heat capacity is unstable thermally.

To calculate Equation (29), we need the analytical forms of $m_h \equiv m(r_h)$ and $T_h \equiv T(r_h)$. Therefore, let us calculate the mass of the BH in an event horizon $r_h$. Thus, we put $\mu(r_h) = 0$, given by Equation (19) and obtain:

$$m_h \underset{Equation\ (19)}{} = \frac{\Lambda_{eff}r_h{}^4 + r_h{}^2 + q^2}{2r_h}.\tag{30}$$

Equation (30) shows that the total mass of BH is a function of $r_h$, the charge and $\Lambda_{eff.}$. For specific values of the charge we plot the relation of the horizon mass-radius in Figure 2a, which shows:

$$m(r_h \to 0) \to \infty, \qquad m(r_h \to \infty) \to \infty.\tag{31}$$

The temperature of BH is calculated at the outer event horizon $r = r_h$ as [93]:

$$T = \frac{\kappa}{2\pi}.\tag{32}$$

Here, $\kappa$ is the surface gravity defined as $\kappa = \frac{\mu'(r_h)}{2}$. The temperatures of the BH (18) is given by:

$$T_h \underset{\text{Equation (19)}}{=} \frac{1}{2\pi}\left(\Lambda_{eff}r_h + \frac{1}{r_h^2}\left[m - \frac{q^2}{r_h}\right]\right),\tag{33}$$

with $T_h$ being the temperature at $r_h$. For our two cases, linear and nonlinear electrodynamics, we depict the temperatures in Figure 2b for specific values. Figure 2b shows that the horizon temperature $T_h$ has a zero value at $r_h = r_{dg}$. However, when $r_h < r_{dg}$, the horizon temperature becomes negative and forms an ultracold black hole. This result was discussed by Davies [94] who said that there are no obvious reasons from the thermodynamical viewpoint that prevent a BH temperature from becoming negative and linked this to a naked singularity. This is exactly what happened in Figure 2b when $r_h < r_{min}$ region. The case of ultracold BH is explained by the existence of a phantom energy field [95], which investigates the decrease of the mass behavior in Figure 2b. When $r_h > r_{dg}$, the temperature becomes positive. When $r_h$ becomes larger, the temperatures of both linear and nonlinear cases change in a similar manner.

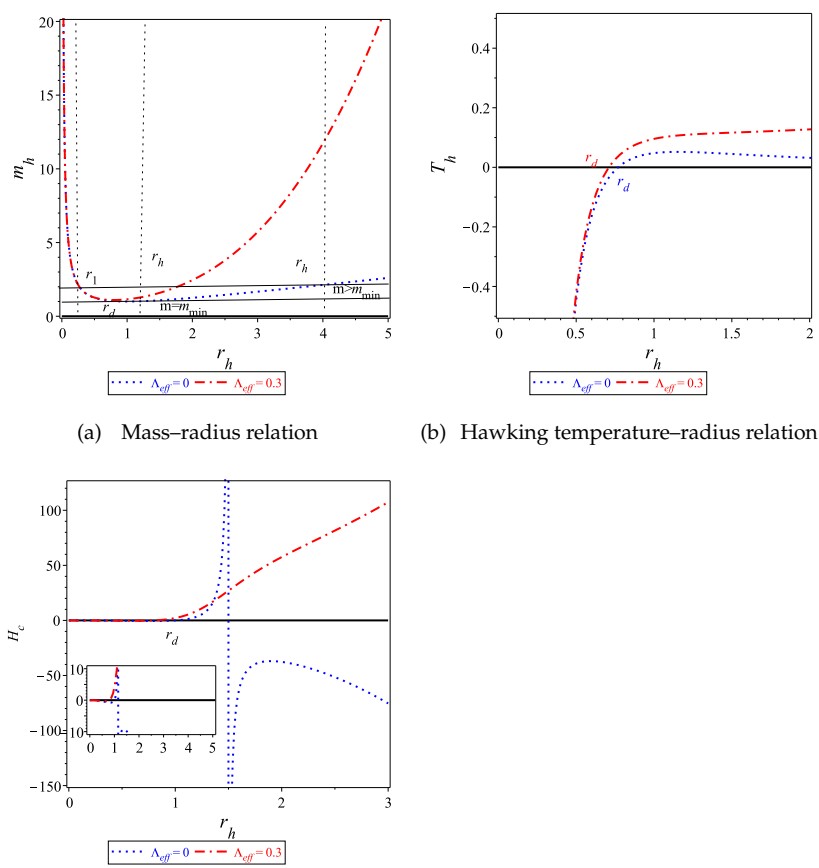

(a) Mass–radius relation

(b) Hawking temperature–radius relation

(c) Heat capacity–radius relation

**Figure 2.** Plots of thermodynamical quantities of BHs. (**a**) The mass-radius relation, which determines the minimal mass. (**b**) The hawking temperature, which vanishes at $r_h$. (**c**) The heat capacity. Moreover, the linear case investigates a second-order phase transition. All the figures are plotted for $m_h = q = 1$.

Now we are going to evaluate the heat capacity, $H_c$. Using Equations (29), (30), and (33) we get:

$$H_{c_{\text{Equation (19)}}} = \frac{2\pi r_h{}^2 \left[3\Lambda_{eff} r_h{}^4 + r_h{}^2 - q^2\right]}{2(\Lambda_{eff} r_h{}^4 + 3q^2 - 2mr_h)}.$$

(34)

The above equation is not easy to obtain from any information; thus, we depicted it in Figure 2c with specific values of the parameters. As shown in Figure 2c, both cases of linear and nonlinear charged BH solutions, $H_c$ vanishes at $r_{dg}$ and also their temperatures. In the GR limit, the linear case, $H_c$ has positive values when $r_h > r_{dg}$; however, when $r_h < r_{dg}$, it has negative values. In the NED case, the heat capacity is always positive unless $r_h < r_{dg}$.

*3.1. First Law of Thermodynamics of the BH Solution (18)*

Using Equation (30) we obtain:

$$M = m_h = \frac{\Lambda_{eff} r_h{}^3}{2} + \frac{r_h}{2} + \frac{q^2}{2r_h}.$$

(35)

Moreover, from the definition of entropy:

$$S = \frac{A}{4} = \pi r_h{}^2,$$

(36)

we can show that the effective cosmological constant and pressure are given as [96]:

$$P = \frac{3\Lambda_{eff}}{8\pi}.$$

(37)

Equation (35) can be rewritten in terms of pressure and entropy as:

$$M(S, q, P) = \frac{1}{6\sqrt{\pi S}}\left(3\pi\, q^2 + 3S + 8P\, S^2\right).$$

(38)

Therefore, the parameters related to $S$, $q$, and $P$ are calculated as:

$$T = \left(\frac{\partial M}{\partial S}\right)_{P,q} = \frac{1}{4\pi r_h}\left(1 - \frac{q^2}{r_h{}^2} + 3\pi r_h{}^2 \Lambda_{eff.}\right),$$

$$\xi = \left(\frac{\partial M}{\partial q}\right)_{S,P} = \frac{q}{r_h}, \qquad V = \left(\frac{\partial M}{\partial P}\right)_{S,q} = \frac{4}{3}\pi r_h{}^3,$$

(39)

where $\xi$, T, and V are the electric potential, temperature, and thermodynamic volume, respectively. Using the above equations, the following Smarr relation is:

$$M = 2TS + \xi q - 2VP,$$

(40)

from which it is easy to prove the first law of thermodynamics as:

$$dM = TdS + \xi\, dq + VdP.$$

(41)

Equation (40) ensures the validity of the first law of the BH (19).

## 4. Discussion and Conclusions

In this research, we have considered spherically symmetric BH in Rastall's theory of gravity. We study the NED spherically symmetric spacetime and derive an exact solution that is affected by the Rastall parameter. This is the first time we derive a NED BH solution

from the field equation of Rastall's gravitational theory. The main contribution of Rastall's parameter in this study comes from the contribution of the trace of the NED, which has a non-vanishing value in contrast to the linear Maxwell theory. We show that the effect of the Rastall parameter acts as a cosmological constant, and the BH behaves asymptotically as (A)dS Reissner–Nordström spacetime. When the Rastall parameter vanishes, we obtaiin spacetime, which asymptotes as flat Reissner–Nordström spacetime.

We have used the geodesic deviation to obtain the stability of the geodesic motion of the NED case. Furthermore, we investigated the horizons and demonstrated that the BHs presented in this study could have two horizons: the event horizon $r_1$ and the effective cosmological one $r_h$. Furthermore, we fixed the minimum value of the BH mass that occurred at the degenerate horizon. We have also studied the thermal phase transitions and showed that in the linear electrodynamics case, i.e., $\epsilon = 0$, the temperature became negative when $r_h < r_d$ and, therefore, heat capacity became negative and, thus, we have unstable BH [97–100]. The same conclusions can be applied to the NED case. However, at $r_h > r_d$, we have a positive value of the $H_c$, which yields a stable BH. Finally, we proved the validity of the first law of thermodynamics. It is worth noting that the result of thermodynamics presented in this study agrees with the study of thermodynamics presented in [101] when the rotation parameter *a* is vanishing.

In this study, we have discussed Rastall's theory using a special form of non-linear electrodynamics. This special form of non-linear electrodynamics reduces in our model to a linear form plus a cosmological constant. However, a deeper analysis is necessary, possibly regarding quantum effects in the universe. Meanwhile, the effects of Rastalls cosmology on the formation and properties of non-linear structures is a very promising research program. Furthermore, the study of $f(R)$-Rastall's theory will be extremely rich in the context of astrophysics [65]. Within the frame of $f(R)$, a BH, which is similar to Reissner–Nordström BH is presented [102] for a specific form of $f(R)$. Is it possible to derive a similar solution within Rastall's $f(R)$? This study will be carried out elsewhere.

**Funding:** This research received no external funding.

**Conflicts of Interest:** The authors declare no conflict of interest.

**Sample Availability:** Samples of the compounds are available from the authors.

## Notes

[1] In this study we assume the relativistic units, i.e., $\chi = \frac{8\pi G}{c^4} = 1$.

[2] In the frame of Rastall theory, Reissner−Nordström is a solution since its Ricci scalar has a vanishing value.

[3] The non-vanishing of the trace is an important property in the frame of Rastall's theory so that the effect of the Rastall parameter may appear unlike Maxwell field theory.

[4] Solution (18) has been checked using Maple software 19.

[5] This result is consistent with what we have done in [57] where the author has shown that the Rastall theory is equivalent to Einstein's general relativity or equivalent to Einstein's field equation plus an arbitrary cosmological constant

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
