# Peer review of "Nonlinear Charged Black Hole Solution in Rastall Gravity"

_universe, doi:10.3390/universe8100510_

Round 1

Reviewer 1 Report

As a minor objection, this referee has a query that the author might want to address before publication. The paper

"Rastall gravity is equivalent to Einstein gravity",

by Matt Visser, arXiv:1711.11500 [gr-qc], leaves this referee wondering if the results presented here could be derived more simply by resorting to said equivalence. A brief explanation (to what extent might the equivalence between Einstein and Rastall gravity affect the results of the paper under review?) would suffice.

Reviewer 2 Report

See the attached pdf file. 

Reviewer 3 Report

The author considers the four-dimensional Rastall gravity theory with the nonlinear electromagnetic field. The author obtains an exact solution with the spherical symmetry in this theory, which coincides with the Reissner-Nordstrom-(A)dS solution and represents a charged black hole geometry with the cosmological constant. Then the author studies the geodesic deviations and the thermodynamics of the obtained solution to investigate its stabilities. I find this work interesting but I recommend the author to consider the following points:

i) Would the solution (18) be a general spherically symmetric solution in this system? Why do the effects by the Rastall gravity and the nonlinear electrodynamics vanish simultaneously in the $c_3=0$ limit? Then the author may discuss some assumptions to obtain an another solution that would not coincide with the Reissner-Nordstrom-(A)dS one.

ii) Since the obtained solution is the Reissner-Nordstrom-(A)dS one, there would be some overlaps between the stability analyses in the sections II-C and III in the present manuscript and the papers arXiv:1902.06783 and arXiv:hep-th/9908022. Then the author may modify these sections to clear the difference between the present discussions and these previous works.

iii) The terms $d\phi ^2$ in the equation (14) and $\sin ^2 d\phi ^2$ below the equation (21) would be replaced with the terms $\sin ^2 \theta d\phi ^2$ and $\sin ^2 \theta d\phi ^2$, respectively.

Round 2

Reviewer 3 Report

I am satisfied with the corrections by the author. Then I recommend the publication of this work.

Author Response

Dear Reviewer,

       There is no comment.

Best

Author